Happy software developers solve problems better: psychological measurements in empirical software engineering

Graziotin Daniel daniel.graziotin@unibz.it
Wang Xiaofeng
Abrahamsson Pekka
Faculty of Computer Science, Free University of Bozen-Bolzano , Bolzano , Italy
Mueller Shane
Electronic publication date: 2014 Mar 11
Publication date: 2014
Volume: 2
Electronic Location ID: e289
Received 2013 Jul 25; Accepted 2014 Feb 4
Copyright: © 2014 Graziotin et al.
Copyright year: 2014
Copyright holder: Graziotin et al.
License: This is an open access article distributed under the terms of the Creative Commons Attribution License, which permits unrestricted use, distribution, and reproduction in any medium, provided the original author and source are credited.
License URL: https://creativecommons.org/licenses/by/3.0/

Keywords: Emotion, Affective state, Software development, Analytical problem-solving, Feeling, Creativity, Mood, Human factors, Human aspects, Affect

Funding: This work was not covered by any funding mechanism.

==============================
For more than thirty years, it has been claimed that a way to improve software developers’ productivity and software quality is to focus on people and to provide incentives to make developers satisfied and happy. This claim has rarely been verified in software engineering research, which faces an additional challenge in comparison to more traditional engineering fields: software development is an intellectual activity and is dominated by often-neglected human factors (called human aspects in software engineering research). Among the many skills required for software development, developers must possess high analytical problem-solving skills and creativity for the software construction process. According to psychology research, affective states—emotions and moods—deeply influence the cognitive processing abilities and performance of workers, including creativity and analytical problem solving. Nonetheless, little research has investigated the correlation between the affective states, creativity, and analytical problem-solving performance of programmers. This article echoes the call to employ psychological measurements in software engineering research. We report a study with 42 participants to investigate the relationship between the affective states, creativity, and analytical problem-solving skills of software developers. The results offer support for the claim that happy developers are indeed better problem solvers in terms of their analytical abilities. The following contributions are made by this study: (1) providing a better understanding of the impact of affective states on the creativity and analytical problem-solving capacities of developers, (2) introducing and validating psychological measurements, theories, and concepts of affective states, creativity, and analytical-problem-solving skills in empirical software engineering, and (3) raising the need for studying the human factors of software engineering by employing a multidisciplinary viewpoint.

Introduction

For more than thirty years, it has been claimed that a way to improve software developers’ productivity and software quality is to focus on people (Boehm & Papaccio, 1988). Some strategies to achieve low-cost but high-quality software involve assigning developers private offices, creating a working environment to support creativity, and providing incentives (Boehm & Papaccio, 1988), in short, making software developers satisfied and happy. Several Silicon Valley companies and software startups seem to follow this advice by providing incentives and perks to make their developers happy (Drell, 2011; Google Inc., 2014; Stangel, 2013) and, allegedly, more productive (Marino & Zabojnik, 2008).

Human factors (called human aspects in software engineering) play an important role in the execution of software processes and the resulting products (Colomo-Palacios et al., 2010; Feldt et al., 2010; Sommerville & Rodden, 1996). This perception of the importance of human aspects in software development, e.g., “Individuals and interactions over processes and tools”, led to the publication of the Agile manifesto (Beck et al., 2001). As noted by Cockburn & Highsmith (2001), “If the people on the project are good enough, they can use almost any process and accomplish their assignment. If they are not good enough, no process will repair their inadequacy—‘people trump process’ is one way to say this.” (p. 131). This claim has received significant attention; however, little evidence has been offered to verify this claim in empirical software engineering research.

The software engineering field faces an additional challenge compared with more traditional engineering fields; software development is substantially more complex than industrial processes. The environment of software development is all but simple and predictable (Dybå, 2000). Much change occurs while software is being developed, and agility is required to adapt and respond to such changes (Williams & Cockburn, 2003). Software development activities are perceived as creative and autonomous (Knobelsdorf & Romeike, 2008). Environmental turbulence requires creativity to make sense of the changing environment, especially in small software organizations (Dybå, 2000). The ability to creatively develop software solutions has been labelled as critical for software firms (Ciborra, 1996; Dybå, 2000) but has been neglected in research.

The software construction process is mainly intellectual (Darcy & Ma, 2005; Glass, Vessey & Conger, 1992). Recently, the discipline of software engineering has begun to adopt a multidisciplinary view and has embraced theories from more established disciplines, such as psychology, organizational research, and human–computer interaction. For example, Feldt et al. (2008) proposed that the human factors of software engineering could be studied empirically by “collecting psychometrics”.1 Although this proposal has begun to gain traction, limited research has been conducted on the role of emotion and mood on software developers’ skills and productivity.

As human beings, we encounter the world through affects; affects enable what matters in our experiences by “indelibly coloring our being in the situation” (Ciborra, 2002, p. 161). Diener et al. (1999) and Lyubomirsky, King & Diener (2005) reported that numerous studies have shown that the happiness of an individual is related to achievement in various life domains, including work achievements. Indeed, emotions play a role in daily jobs; emotions pervade organizations, relationships between workers, deadlines, work motivation, sense-making and human-resource processes (Barsade & Gibson, 2007). Although emotions have been historically neglected in studies of industrial and organizational psychology (Muchinsky, 2000), an interest in the role of affect on job outcomes has increased over the past decade (Fisher & Ashkanasy, 2000). The relationship between affect on the job and work-related achievements, including performance (Barsade & Gibson, 2007; Miner & Glomb, 2010; Shockley et al., 2012) and problem-solving processes, such as creativity, (Amabile et al., 2005; Amabile, 1996) has been of interest for recent research.

Despite the fact that the ability to sense the moods and emotions of software developers may be essential for the success of an Information Technology firm (Denning, 2012), software engineering research lacks an understanding of the role of emotions in the software development process (Khan, Brinkman & Hierons, 2010; Shaw, 2004). In software engineering research, the affective states of software developers have been investigated rarely in spite of the fact that affective states have been a subject of other Computer Science disciplines, such as human–computer interaction and computational intelligence (Lewis, Dontcheva & Gerber, 2011; Tsonos, Ikospentaki & Kouroupetrolgou, 2008). Thus, we believe that studying the affective states of software developers may provide new insights about ways to improve overall productivity.

Many of the tasks that software developers engage in require problem solving. For example, software developers need to plan strategies to find a possible solution to a given problem or to generate multiple creative and innovative ideas. Therefore, among the many skills required for software development, developers need to possess high analytical problem-solving skills and creativity. Both of these are cognitive processing abilities. Indeed, software development activities are typically not physical. Software development is complex and intellectual (Darcy & Ma, 2005; Glass, Vessey & Conger, 1992), and it is accomplished through cognitive processing abilities (Fischer, 1987; Khan, Brinkman & Hierons, 2010). Some cognitive processes have been shown to be deeply linked to the affective states of individuals (Ilies & Judge, 2002). Furthermore, to the best of our knowledge, the relationship between affective states and the creativity and analytical problem-solving skills of software developers in general has never been investigated.

This article offers several contributions: (1) it provides a better understanding of the impact of affective states on the creativity and analytical problem-solving capacities of developers; (2) it introduces and validates psychological measurements, theories, and concepts of affective states, creativity and analytical problem-solving skills in empirical software engineering; and (3) it raises the need to study human factors in software engineering by employing a multidisciplinary viewpoint.

Next, we will review some of the background research on how affective states impact creative problem solving.2 Following the background section, we will report a new experiment that establishes the relationship between affect and productivity in software developers.

Affective states

In general, affective states has been defined to as “any type of emotional state …often used in situations where emotions dominate the person’s awareness” (VandenBos, 2013). However, the term has been employed more generally to mean emotions and moods. Many authors have considered mood and emotion to be interchangeable terms (Baas, De Dreu & Nijstad, 2008; Schwarz & Clore, 1983; Schwarz, 1990; Wegge et al., 2006), but it has been acknowledged that numerous attempts exist to differentiate these terms (Wegge et al., 2006; Weiss & Cropanzano, 1996). For example, it has been suggested that a difference between moods and emotions lies in an absence of a causal factor in the phenomenal experience of the mood (Weiss & Cropanzano, 1996). According to several authors, emotions and moods are affective states (Fisher, 2000; Khan, Brinkman & Hierons, 2010; Oswald, Proto & Sgroi, 2008; Weiss & Cropanzano, 1996). It has also been argued that a distinction is not necessary for studying cognitive responses that are not strictly connected to the origin of the mood or emotion (Weiss & Cropanzano, 1996). For the purposes of this investigation, we have adopted the same stance and employed the noun affective states as an umbrella term for emotions and moods.

Measuring affective states

Psychology studies have often categorized affective states in terms of negative, (occasionally) neutral, and positive affective states. In the case of controlled experiments, grouping is usually based on manipulations that induce affective states. Several techniques have been employed to induce affective states on participants, such as showing films, playing certain types of music, showing pictures and photographs, or allowing participants to remember happy and sad events in their lives (Lewis, Dontcheva & Gerber, 2011; Westermann & Spies, 1996). However, recent studies have questioned the effects of mood-induction techniques, especially when studying the pre-existing affective states of participants (Forgeard, 2011). Alternately, some studies have used quasi-experimental designs that select participants with various affective states, which have usually been based on answers to questionnaires.

One of the most notable measurement instruments for affective states is the Positive and Negative Affect Schedule (PANAS; Watson, Clark & Tellegen, 1988). The PANAS is a 20-item survey that represents positive affects (PA) and negative affects (NA). However, several criticisms have been made regarding this instrument. The PANAS reportedly omits core emotions such as bad and joy while including items that are not considered emotions, such as strong, alert, and determined (Diener et al., 2009a; Li, Bai & Wang, 2013). Others have argued that the PANAS is not sensitive to or inclusive of cultural differences in the desirability of emotions (Li, Bai & Wang, 2013; Tsai, Knutson & Fung, 2006). Furthermore, a considerable redundancy has been found in the items of the PANAS (Crawford & Henry, 2004; Li, Bai & Wang, 2013; Thompson, 2007). The PANAS has also been reported to capture only high-arousal feelings in general (Diener et al., 2009a).

Recently, scales have been proposed that reduce the number of the PANAS scale items and that overcome some of its shortcomings. Diener et al. (2009a) developed the Scale of Positive and Negative Experience (SPANE). The SPANE assesses a broad range of pleasant and unpleasant emotions by asking participants to report their emotions in terms of the frequency of the emotion during the last four weeks. The SPANE is a 12-item scale that is divided into two subscales (SPANE-P and SPANE-N) that assess positive and negative affective states. The answers to the items are given on a five-point scale ranging from one (very rarely or never) to five (very often or always). For example, a score of five for the joyful item means that the respondent experienced this affective state very often or always during the last four weeks. The SPANE-P and SPANE-N scores are the sum of the scores given to their respective six items; thus, the scores range from six to thirty. The two scores can be further combined by subtracting the SPANE-N from the SPANE-P, which results in the Affect Balance Score (SPANE-B). The SPANE-B is an indicator of the pleasant and unpleasant affective states caused by how often positive and negative affective states have been felt by the participant. The SPANE-B ranges from −24 (completely negative) to +24 (completely positive).

The SPANE measurement instrument has been reported to be capable of measuring positive and negative affective states regardless of their sources, arousal level or cultural context, and it captures feelings from the emotion circumplex (Diener et al., 2009a; Li, Bai & Wang, 2013). The timespan of four weeks was chosen in the SPANE to provide a balance between the sampling adequacy of feelings and the accuracy of memory (Li, Bai & Wang, 2013) and to decrease the ambiguity of people’s understanding of the scale itself (Diener et al., 2009a). The SPANE has been validated to substantially converge to other affective states measurement instruments, including the PANAS (Diener et al., 2009a). The scale provided good psychometric properties in the introductory research (Diener et al., 2009a) and in numerous follow-ups, which included up to twenty-one thousand participants in a single study (Dogan, Totan & Sapmaz, 2013; Li, Bai & Wang, 2013; Silva & Caetano, 2011). Additionally, the scale provided consistency across full-time workers and students (Silva & Caetano, 2011).

Even if the SPANE-B provides a graded scale rather than a categorical scale, it could be employed to split participants into groups using a median split. It is common to adopt the split technique on affective states measures (Berna et al., 2010; Forgeard, 2011; Hughes & Stoney, 2000).

Affective states and software developers

Several past research efforts have examined the role of affective states on software developers. For example, Shaw (2004) observed that the role of emotions in the workplace has been the subject of study management research, but information systems research has focused on job outcomes such as stress, turnover, burnout, and satisfaction. The study explored the emotions of information technology professionals and how these emotions can help explain their job outcomes. The paper employed the Affective Events Theory (Weiss & Cropanzano, 1996) as a framework for studying the fluctuation of the affective states of 12 senior-level undergraduate students who were engaged in a semester-long implementation project for an information systems course. The participants were asked to rate their affective states during or immediately after their episodes of work on their project. At four intervals during the project, they filled out a survey on stress, burnout, emotional labor, and identification with their teams. Shaw considered each student to be a single case study because a statistical analysis was not considered suitable. The study showed that the affective states of a software developer may dramatically change during a period of 48 h and that the Affective Events Theory proved its usefulness in studying the affective states of software developers while they work. Shaw (2004) concluded by calling for additional research.

This call was echoed by Khan, Brinkman & Hierons (2010). In their study, a correlation with cognitive processing abilities and software development was demonstrated theoretically. The authors constructed a theoretical two-dimensional mapping framework in two steps. The authors reported two empirical studies on affective states and software development. The studies were related to the impact of affective states on developers’ debugging performance. In the first study, affective states were induced to the software developers. Subsequently, the programmers completed a quiz on software debugging. In the second study, the participants were asked to write a trace on paper of the execution of algorithms implemented in Java. After 16 min of algorithm tracing, arousal was induced in the participants. Subsequently, the participants continued their debugging task. The overall study provided empirical evidence for a positive correlation between the affective states of software developers and their debugging performance.

Finally, Graziotin, Wang & Abrahamsson (2013) conducted a correlational study on the affective states of developers and their self-assessed productivity while constructing software. The research employed the dimensional view of affective states and included a pictorial survey to assess the affective states raised by the software development task. The study observed eight developers working on their individual software projects. Their affective states and their self-assessed productivity were measured in intervals of 10 min. The analysis of the correlation employed a linear mixed-effects model. Evidence was found that valence and dominance towards a software development task are positively correlated with the self-assessed productivity of developers.

Problem-solving performance and affective states

Researchers have sometimes distinguished between two modes of creative and analytic problem solving: convergent and divergent thinking (Cropley, 2006; Csikszentmihalyi, 1997), which map roughly onto creativity and analytic problem solving studies, according to Csikszentmihalyi (1997). Divergent thinking leads to no agreed-upon solutions and involves the ability to generate a large quantity of ideas that are not necessarily correlated (Csikszentmihalyi, 1997). Convergent thinking involves solving well-defined, rational problems that often have a unique, correct answer and emphasizes speed and working from what is already known, which leaves little room for creativity because the answers are either right or wrong (Cropley, 2006; Csikszentmihalyi, 1997).

Past research has found mixed evidence regarding the relationships between positive or negative affective states and problem solving performance. According to a recent meta-analysis on the impact of affective states on creativity (in terms of creative outcomes), positive affective states lead to a higher quality of generated ideas than do neutral affective states, but there are no significant differences between negative and neutral affective states or between positive and negative affective states (Baas, De Dreu & Nijstad, 2008). Another recent meta-analysis agreed that positive affective states have moderately positive effects on creativity in comparison to neutral affective states. However, this study showed that positive affective states also have weakly positive effects on creativity in comparison to negative affective states (Davis, 2009). Similarly, Lewis, Dontcheva & Gerber (2011) provided evidence for higher creativity under induced positive and negative affective states, in comparison to non-induced affective states. Forgeard (2011) showed that participants who were low in depression possessed higher creativity when negative affective states were induced, and no benefits were found in the participants when positive affective states were induced. Sowden & Dawson (2011) found that the quantity of generated creative ideas is boosted under positive affective states, but no difference in terms of quality was found in their study. However, studies have demonstrated that negative affective states increase creativity (George & Zhou, 2002; Kaufmann & Vosburg, 1997). As argued by Fong (2006), no clear relationship has been established between affective states and problem solving creativity. No direction could be predicted on a difference between the creativity and affective states of software developers.

In contrast to the case for creativity, fewer studies have investigated how affective states influence analytic problem solving performance. The understanding of the relationship is still limited even in psychology studies. In her literature review on affects and problem-solving skills, Abele-Brehm (1992) reported that there is evidence that negative affects foster critical and analytical thinking. Successive theoretical contributions have been in line with this suggestion. In their mood-as-information theoretical view, Schwarz & Clore (2003) argued that negative affects foster a systematic processing style characterized by bottom-up processing and attention to the details, and limited creativity. Spering, Wagener & Funke (2005) observed that negative affects promoted attention to the details to their participants, as well as analytical reasoning. It appears that analytical problem-solving skills—related to convergent thinking—are more influenced by negative affective states than by positive affective states. However, there are studies in conflict with this stance. Kaufmann & Vosburg (1997) reported no correlation between analytical problem-solving skills and the affective states of their participants. On the other hand, the processes of transferring and learning analytical problems have been reported to deteriorate when individuals are experiencing negative emotions (Brand, Reimer & Opwis, 2007). Melton (1995) observed that individuals feeling positive affects performed significantly worse on a set of syllogisms (i.e., logical and analytical reasoning). Consequently, based on the limited studies, no clear prediction about the relationship between affective states and analytic problem solving skill could be made.

Because of the lack of a clear relationship between affective states and problem-solving performance, we designed an experiment to test two related high-level hypotheses. We hypothesize that affective states will impact (1) the creative work produced by software developers and (2) their analytic problem-solving skills.

To test the hypotheses we obtained various measures of creativity, and we developed a measure of analytical problem-solving. Often, a creative performance has been conceptualized in terms of the outcome of the process that leads to the creation of the creative results (Amabile, 1982; Davis, 2009). A widely adopted task asks individuals to generate creative ideas for uncommon and bizarre problems (Forgeard, 2011; Kaufman et al., 2007; Lewis, Dontcheva & Gerber, 2011; Sowden & Dawson, 2011). For assessing the creativity of our participants, we used a “caption-generating” task. The quality of the creative outcome was assessed with subjective ratings by independent judges, and the quantity of the generated captions was recorded.

A common approach for testing analytical problem-solving is to assign points to the solution of analytical tasks (Abele-Brehm, 1992; Melton, 1995). We used the Tower of London test (Shallice, 1982), a game designed to assess planning and analytical problem-solving. The Tower of London game is a very high-level task that resembles algorithm design and execution. This task reduced the limitations that would have been imposed by employing a particular programming language. Furthermore, such a level of abstraction permits a higher level of generalization because the results are not bound to a particular programming language.

To our knowledge, there have been no studies in software engineering research using software development tasks that are suitable for measuring the creativity and analytical problem-solving skills of software developers. Although strict development tasks could be prepared, there would be several threats to validity. Participants with various backgrounds and skills are expected, and it is almost impossible to develop a software development task suitable and equally challenging for first year BSc students and second year MSc students. The present study remained at a higher level of abstraction. Consequently, creativity and analytical problem-solving skills were measured with validated tasks from psychology research.

Materials and Methods

Participant characteristics

Forty-two student participants were recruited from the Faculty of Computer Science at the Free University of Bozen-Bolzano. There were no restrictions in the gender, age, nationality, or level of studies of the participants. Participation was voluntary and given in exchange for research credits. The affective states of the participants were natural, i.e., random for the researchers. Of the 42 participants, 33 were male and nine were female. The participants had a mean age of 21.50 years old (standard deviation (SD) = 3.01 years) and were diverse in nationality: Italian 74%, Lithuanian 10%, German 5%, and Ghanaian, Nigerian, Moldavian, Peruvian, or American, with a 2.2% frequency for each of these latter nationalities. The participants’ experience in terms of years of study was recorded (M = 2.26 years, SD = 1.38).

Institutional review board approval for conducting empirical studies on human participants was not required by the institution. However, written consent was obtained from all of the subjects. The participants were advised, both informally and on the consent form, about the data retained and that anonymity was fully ensured. No sensitive data were collected in this study. The participants were assigned a random participant code to link the gathered data. The code was in no way linked to any information that would reveal a participant’s identity.

All of the students participated in the affective states measurement sessions and the two experimental tasks. However, the results of one participant from the creativity task and another from the analytical problem-solving task have been excluded; the two participants did not follow the instructions and submitted incomplete data. Therefore, the sample size for the two experiment tasks was N = 41. None of the participants reported previous experience with the tasks.

Materials

For the two affective states measurement sessions, the participants completed the SPANE questionnaire through a Web-based form, which included the related instructions. The SPANE questionnaire instructions that were provided to the participants are available in the article by Diener et al. (2009a) and are currently freely accessible on one of the author’s academic website (Diener et al., 2009b).

Six color photographs with ambiguous meanings were required for the creativity task. Figure 1 displays one of the six photographs. For legal reasons, the photographs are available from the authors upon request only.

Figure 1 A photograph for the creativity task.

“Untitled - London ’11” by i.witness. Copyright © 2011 i.witness. Reproduced here with kind permission from the author. Available from http://www.flickr.com/photos/i_witness/6587622327/in/photostream/

For the analytical problem-solving task, a version of the Tower of London task implemented in the open source Psychology Experiment Building Language (PEBL; Mueller & Piper, 2014; Mueller, 2012) that has been used previously to examine age-related changes in planning and executive function (Piper et al., 2011) was used to assess analytic problem solving. The PEBL instructed the participants, provided the task, and collected several metrics, including those of interest for our study. One computer per participant was required.

Procedure

The experimental procedure was composed of four activities: (1) the affective states measurement (SPANE), (2) the creativity task, (3) the affective states measurement (SPANE), and (4) the analytical problem-solving task. The second affective states measurement session was conducted to limit the threats to validity because the first task may provoke a change in the affective states of the participants.

The participants arrived for the study knowing only that they would be participating in an experiment. As soon as they sat at their workstation, they read a reference sheet, which is included in Article S1. The sheet provided a summary of all of the steps of the experiment. The researchers also assisted the participants during each stage of the experiment. The participants were not allowed to interact with each other.

During the creativity task, the participants received two random photographs from the set of the six available photographs, one at a time. The participants imagined participating in the Best Caption of the Year contest and tried to win the contest by writing the best captions possible for the two photographs. They wrote as many captions as they wanted for the pictures. The creativity task instructions are available as an appendix in the study by Forgeard (2011).

During the analytical problem-solving task, the participants opened the PEBL software. The software was set up to automatically display the Tower of London game, namely the Shallice test ([1, 2, 3] pile heights, 3 disks, and Shallice’s 12 problems). The PEBL software displayed the instructions before the start of the task. The instructions stated how the game works and that the participants had to think about the solution before starting the task, i.e., making the first mouse click. Figure 2 provides a screenshot of the first level of the game. Because PEBL is open-source software, the reader is advised to obtain the PEBL software to read the instructions.

Figure 2 The first level of the Tower of London game.

Although the participants did not have strict time restrictions for completing the tasks, they were advised of the time usually required to complete each task and that the second task would begin only after every participant finished the first task.

The participants were not aware of any experimental settings nor of any purpose of the experiment.

Two supervisors were present during the experiment to check the progress of the participants and to answer their questions. All of the steps of the experiment were automated with the use of a computer, except for the caption production in the creativity task. The captions were manually transcribed in a spreadsheet file. For this reason, a third person double checked the spreadsheet containing the transcribed captions.

The study was conducted in January 2012. The designed data collection process was followed fully. No deviations occurred. Each of the tasks required 30 min to be completed, and the participants completed the two surveys in 10 min each. No participants dropped from the study.

Measures

To measure creativity according to the Consensual Assessment Technique (Amabile, 1982), independent judges who are experts in the field of creativity scored the captions using a Likert-item related to the creativity of the artifact to be evaluated. The judges had to use their own definition of creativity (Amabile, 1982; Kaufman et al., 2007). The Likert-item is represented by the following sentence: This caption is creative. The value associated to the item ranges from one (I strongly disagree) to seven (I strongly agree). The judges were blind to the design and the scope of the experiment. That is, they received the six pictures with all of the participants’ captions grouped per picture. The judges were not aware of the presence of other judges and rated the captions independently. Ten independent judges were contacted to rate the captions produced in the creativity task. Seven judges responded, and five of the judges completed the evaluation of the captions. These five judges included two professors of Design & Arts, two professors of humanistic studies, and one professor of creative writing.

The present study adopted measurements of quality and quantity for the assessment of creativity. The quality dimension of creativity was measured by two scores. The first quality score was the average of the scores assigned to all of the generated ideas of a participant (ACR). The second quality score was the best score obtained by each participant (BCR), as suggested by Forgeard (2011) because creators are often judged by their best work rather than the average of all of their works (Kaufman et al., 2007). The quantity dimension was represented by the number of generated ideas (NCR), as suggested by Sowden & Dawson (2011).

Measuring analytical problem-solving skills is less problematic than measuring creativity. There is only one solution to a given problem (Cropley, 2006). The common approach in research has been to assign points to the solution of analytical tasks (Abele-Brehm, 1992; Melton, 1995). This study employed this approach to combine measures of quality and quantity by assigning points to the achievements of analytical tasks and by measuring the time spent on planning the solution. The Tower of London game (a.k.a. Shallice’s test) is a game aimed to determine impairments in planning and executing solutions to analytical problems (Shallice, 1982). It is similar to the more famous Tower of Hanoi game in its execution. Figure 2 provides a screenshot of the game. The rationale for the employment of this task is straightforward.

The Analytical Problem Solving (APS) score is defined as the ratio between the progress score achieved in each trial of the Tower of London Game (TOLSS) and the number of seconds needed to plan the solution to solve each trial (PTS). The TOLSS scores range from 0 to 36 because there are 12 problems to be solved and each one can be solved in a maximum of three trials. PTS is the number of milliseconds that occurred between the presentation of the problem and the first mouse click in the program. To have comparable results, a function to map the APS ratio to a range from 0.00 to 1.00 was employed.

Results

The data were aggregated and analyzed using the open-source R software (R Core Team, 2013). The SPANE-B value obtained from this measurement session allowed us to estimate the SPANE-B population mean for software developers, µSPANE-B-DEV = 7.58, 95% CI [5.29, 9.85]. The median value for the SPANE-B was nine. This result has consequences in the discussion of our results which we offer in the next section.

The multiple linear and polynomial regression analyses on the continuous values for the various SPANE scores and the task scores did not yield significant results. Therefore, the data analysis was performed by forming two groups via a median split of the SPANE-B score. The two groups were called N-POS (for non-positive) and POS (for positive). Before the creativity task, 20 students were classified as N-POS and 21 students were classified as POS.

The histograms related to the affective state distributions and the group compositions have been included as supplemental files of this article (Figs. S1 and S7). These data are not crucial for the purposes of this investigation. However, they have been attached to this article for the sake of completeness. The same holds for the boxplots and the scatterplots representing non-significant data.

Table 1 Mean and standard deviation of the task scores divided by the groups.

	N-POS	POS	
Variable	M (SD)	95% CI	M (SD)	95% CI	
ACR	3.13 (0.45)	[2.92, 3.35]	3.08 (0.58)	[2.81, 3.35]	
BCR	4.02 (0.76)	[3.67, 4.38]	3.98 (0.76)	[3.63, 4.32]	
NCR	4.70 (2.34)	[3.60, 5.50]	5.90 (3.46)	[4.00, 7.50]	
APS	0.14 (0.04)	[0.12, 0.17]	0.20 (0.08)	[0.17 0.25]	
Notes.

ACR the average of the scores assigned to all of the generated ideas of a participant

BCR the best score obtained by each participant

NCR the number of generated ideas

APS the analytical problem-solving score

N-POS non-positive group

POS positive group

Table 1 summarizes the task scores of the two groups for the two tasks. The two creativity scores of ACR and BCR showed many commonalities. Visual inspections of the scatterplots of the ACR (Fig. S5) and BCR (Fig. S6) scores versus the SPANE-B score suggested a weak trend of higher creativity when the SPANE-B value tended to its extreme values (−24 and +24). The median for the number of generated captions (NCR) was four for the N-POS group and six for the POS group. However, the lower quartiles of the two groups were almost the same, and there was a tiny difference between the two upper quartiles (Fig. S4).

We hypothesized that affective states would impact the creative work produced by software developers, without a direction of such impact. The hypothesis was tested using unpaired, two-tailed t-tests. There was no significant difference between the N-POS and POS groups on the BCR score (t(39) = 0.20, p > .05, d = 0.07, 95% CI [−0.43, 0.53]) or the ACR score (t(39) = 0.31, p > .05, d = 0.10, 95% CI [−0.28, 0.38]). The third test, which regarded the quantity of generated creative ideas (NCR), required a Mann–Whitney U test because the assumptions of normality were not met (Shapiro–Wilk test for normality, W = 0.89, p = 0.02 for N-POS and W = 0.87, p = 0.01 for POS). There was no significant difference between the N-POS and POS groups on the NCR score (W = 167.50, p > .05, d = −0.41, 95% CI [−2.00, 1.00]).

The second SPANE questionnaire session was performed immediately after the participants finished the creativity task. The average value of the SPANE-B was M = 8.70 (SD = 6.68), and the median value was 10. There was a significant increase in the SPANE-B value of 1.02 (t(39) = 3.00, p < 0.01, d = 0.96, 95% CI [0.34, 1.71]). Therefore, a slight change in the group composition occurred, with 19 students comprising the N-POS group and 22 students comprising the POS group. Cronbach (1951) developed the α as a coefficient of internal consistency and interrelatedness especially designed for psychological tests. The value of Cronbach’s α ranges from 0.00 to 1.00, where values near 1.00 indicate excellent consistency (Cortina, 1993; Cronbach, 1951). The Cronbach’s α reliability measurement for the two SPANE questionnaire sessions was α = 0.97 (95% CI [0.96, 0.98]), which indicates excellent consistency. We discuss the consequences of these results in the next section.

Figure 3 Boxplots for the analytical problem-solving (APS) of the N-POS and POS groups.

Figure 4 Scatterplot for the analytical problem-solving (APS) vs. the affect balance (SPANE-B) between the N-POS and POS groups.

We hypothesized that affective states would impact the analytic problem-solving skills of software developers. The boxplots for the APS score in Fig. 33 suggest a difference between the two groups, and the relevant scatterplot in Fig. 4 suggested that the APS points for the N-POS group may be linear and negatively correlated with the SPANE-B; excellent APS score were achieved only in the POS group. The hypothesis was tested using an unpaired, two tailed t-test with Welch’s correction because a significant difference in the variances of the two groups was found (F-test for differences in variances, F(21, 18) = 3.32, p = 0.01, 95% CI [1.30, 8.17]). There was significant difference between the N-POS and POS groups on the APS score (t(33.45) = −2.82, p = 0.008, d = −0.91, 95% CI [−0.11, −0.02]). A two-sample permutation test confirmed the results (t(168), p = 0.01, CI [−13.19, −1.91]).

Discussion

Our first SPANE measurement session offered the estimation µSPANE-B-DEV = 7.58 (95% CI [5.29, 9.85]) for the population’s true mean. That is, it might be that the central value for the SPANE-B for software developers is above seven and significantly different from the central value of the measurement instrument, which is zero. While we further reflect on this in the Limitations section, the reader should note that our discussion of the results takes this into account, especially when we compare our results with related work.

The empirical data did not support a difference in creativity with respect to the affective states of software developers in terms of any of the creativity measures we used. The results of this study agree with those of Sowden & Dawson (2011), who did not find a difference in the creativity of the generated ideas with respect to the affective states of the participants. We found no significant difference in the number of creative ideas generated, which is in contrast to Sowden & Dawson (2011), who found that participants in the positive condition produced more solutions than did those in the neutral and negative conditions. Instead, the results of this study deviate from those in the study by Forgeard (2011), where non-depressed participants provided more creative captions under negative affective states. Nevertheless, it must be noted that the depression factor has not been controlled in this study. Overall, the results of this study contrast with past research that places affects—regardless of their polarity and intensity—as important contributors of the creative performance of individuals.

As we reported in the previous section, the second SPANE session was included for limiting the threats to validity because the first task could provoke a change in the affective states of the participants. During the execution of the creativity task, we observed how the participants enjoyed the task and how happily they committed to the task. This observation was mirrored by the data; the participants generated 220 captions, averaging 5.24 captions per participant. This enjoyment of the first task was reflected by the second SPANE measurement session, as there was a significant increase in the SPANE-B value of 1.02 (t(39) = 3.00, p < 0.01, d = 0.96, 95% CI [0.34, 1.71]). This further validates the capabilities of the adopted measurement instrument for the affective state measurements and shows that even simple and short activities may impact the affective states of software developers. The Cronbach’s α value of 0.97 of the two SPANE measurement sessions present evidence that the participants provided stable and consistent data. The choice to include a second affective states measurement session in the design of the study is justified by the obtained results.

The empirical data supported a difference in the analytical problem-solving skills of software developers regarding their affective states. More specifically, the results suggest that the happiest software developers are more productive in analytical problem solving performance. The results of this study contrast with the past theoretical contributions indicating that negative affective states foster analytic problem-solving performance (Abele-Brehm, 1992; Schwarz & Clore, 2003; Spering, Wagener & Funke, 2005). The results of this study are in contradiction to those obtained by Melton (1995), who observed that individuals feeling positive affects performed significantly worse on a set of syllogisms (i.e., logical and analytical reasoning). Although we adopted rather different tasks, our participants feeling more positive affects performed significantly better than any other participants. Likewise, our results are in contradiction to those of Kaufmann & Vosburg (1997), where the performance on the analytic task was negatively related to anxiety (both trait and state) of the participants. However, there was no significant relationship between either positive or negative mood of the participants and their analytical problem-solving performance. Yet, our results tell that happiest software developers outperformed all the other participants in terms of analytic problem-solving.

Limitations

The primary limitation of this study lies in the sample; the participants were all Computer Science students. Although there is diversity in the nationality and experience in years of study of the participants, they have limited software development experience compared with professionals. However, Kitchenham et al. (2002) and Tichy (2000) argued that students are the next generation of software professionals. Thus, they are remarkably close to the population of interest and may even be more updated on the new technologies (Kitchenham et al., 2002; Tichy, 2000). Höst, Regnell & Wohlin (2000) found non-significant differences in the performance of professional software developers and students on the factors affecting the lead-time of projects. There is an awareness that not all universities offer the same curricula and teaching methods and that students may have various levels of knowledge and skills (Berander, 2004). Still, given the high level of abstraction provided by the tasks in this study, a hypothetical difference between this study’s participants and software professionals would likely be in the magnitude and not in the direction of the results (Tichy, 2000). Lastly, the employed affective states measurement instrument, SPANE, provided consistent data across full-time workers and students (Silva & Caetano, 2011).

Another limitation is that full coverage of the SPANE-B range in the negative direction could not be obtained. Although 42 participants were recruited, the SPANE-B score did not fall below the value of minus nine, and its average value was always greater than +7 on a scale of [−24, +24]. Before the experiment, a more homogeneous distribution of participants was expected for the SPANE-B score. However, there is actually no evidence that the distribution of SPANE-B scores for the population of software developers should cover the full range of [−24, +24]. Additionally, studies estimating the SPANE-B mean for any population are not known. For this reason, an estimation of the affective states population mean for software developers was offered by this study: µSPANE-B-DEV = 7.58, 95% CI [5.29, 9.85]. Thus, it may be that the population’s true mean for the SPANE-B is above +7 and significantly different from the central value of the measurement instrument. This translated to a higher relativity when we discussed our results, especially for the comparison with related work. However, the results of this study are not affected by this discrepancy.

A third limitation lies in the employment of a median split to compose the groups. Employing a median split removed the precision that would have been available in a continuous measure of the SPANE-B.4 Despite this, using a median split was necessary because no known regression technique could yield valid results; median splits on affective state measurements are not uncommon in similar research (Berna et al., 2010; Forgeard, 2011; Hughes & Stoney, 2000).

Implications and future research

The theoretical implications of this study are that positive affective states of software developers are indicators of higher analytical problem-solving skills. Although the same is not shown for creativity, the data trends offer inspiration to continue this avenue of study. An implication for research in software engineering is that the study of affective states of the various stakeholders involved in the process of software construction should be taken into account and should become an essential part of the research in the field.

The results have implications for management styles and offer an initial support for the claim that an increase in productivity is expected by making software developers happy. The results may partially justify the workplace settings of currently successful and notable Silicon Valley ventures, which provide several incentives to entertain their software developers (Drell, 2011; Stangel, 2013). However, if the results were generalized we would suspect that creative problem solving will not be impacted in general but analytic skills might be.

Future research should provide additional details for the claims reported in this article. A replication of this experiment with a larger order of magnitude may provide significant data and could even enable regression analyses to verify how the intensity of affective states may impact the creativity of software developers. It is necessary to study the affective states of software developers from a process-oriented view to observe a possible correlation with work-related achievements and productivity while developing software. Qualitative research should explain how the creativity of software developers influences design artifacts and the source-code of a software system. Research can be conducted on how mood induction effects may affect the quality of a software system and the productivity of a developer.

Conclusions

For decades, it has been claimed that a way to improve software developers’ productivity and software quality is to focus on people and to make software developers satisfied and happy. Several Silicon Valley companies and software startups are following this advice, by providing incentives and perks, to make developers happy. However, limited research has supported such claim.

A proposal to study human factors in empirical software engineering research has been to adopt psychological measurements. By observing the reference fields—primarily psychology and organizational research—we understood that software developers solve problems in creative and analytic ways through cognitive processing abilities. Cognitive processing abilities are linked deeply with the affective states of individuals, i.e., emotions and moods.

This paper reported a study—built on the acquired multidisciplinary knowledge—on the importance of affective states on crucial software development skills and capacities, namely analytical problem-solving skills (convergent thinking) and creativity (divergent thinking). It has been shown that happiest software developers are significantly better analytical problem solvers. Although the same could not be shown for creativity, more research on this matter is needed.

The understanding provided by this study should be part of basic science—i.e., essential—in software engineering research, rather than leading to direct, applicable results. This work (1) provides a better understanding of the impact of the affective states on the creativity and analytical problem-solving capacities of developers, (2) introduces and validates psychological measurements, theories, and concepts of affective states, creativity and analytical-problem-solving skills in empirical software engineering and (3) raises the need to study human factors in software engineering by employing a multidisciplinary viewpoint.

Although the claim people trump process is far from being empirically validated, this study provides tools, evidence, and an attitude towards its validation. This study calls for further research on the affective states of software developers.

Software developers are unique human beings. By embracing a multidisciplinary view, human factors in software engineering can be effectively studied. By inspecting how cognitive activities influence the performance of software engineers, research will open up a completely new angle and a better understanding of the creative activity of the software construction process.

Supplemental Information

Article S1 Reference sheet for the experiment participants

Click here for additional data file.

Figure S1 Histogram of the affect balance (SPANE-B) before the creativity task

Click here for additional data file.

Figure S2 Boxplots for the average creativity (ACR) of the N-POS and POS groups

Click here for additional data file.

Figure S3 Boxplots for the best creativity (BCR) of the N-POS and POS groups

Click here for additional data file.

Figure S4 Boxplots for the number of creative ideas (NCR) of the N-POS and POS groups

Click here for additional data file.

Figure S5 Scatterplot for the average creativity (ACR) vs. the affect balance (SPANE-B) between the N-POS and POS groups

Click here for additional data file.

Figure S6 Scatterplot for the best creativity (BCR) vs. the affect balance (SPANE-B) between the N-POS and POS groups

Click here for additional data file.

Figure S7 Histogram of the affect balance (SPANE-B) before the analytical problem-solving task

Click here for additional data file.

The authors would like to thank the students who participated in the experiment. The authors would also like to acknowledge Elena Borgogno, Cristiano Cumer, Federica Cumer, Kyriaki Kalimeri, Paolo Massa, Matteo Moretti, Maurizio Napolitano, Nattakarn Phaphoom, and Juha Rikkilä for their kind help during this study. Last but not least, the authors are grateful for the insightful comments and understanding offered by the Academic Editor Shane T. Mueller and two anonymous reviewers.

Additional Information and Declarations

Competing Interests

Author Contributions

Human Ethics

1 The software engineering literature has sometimes used the term psychometrics to describe general psychological measures that might be used along with other software development metrics. However, psychometrics has a specific meaning within psychological research and involves establishing the reliability and validity of a psychological measurement. In this article, we use the more appropriate term of psychological measurement to refer to this concept.

2 It is an objective of this manuscript to bring concepts, theories, and measurements from psychology to the body of knowledge of software engineering. Therefore, some information provided in this article—especially in the Introduction—may appear redundant and obvious for a reader acquainted with psychology.

3 The color scheme for the graphs of this study have been generated by following the guidelines for producing colorblind-friendly graphics (Okabe & Ito, 2008).

4 The authors are thankful to an anonymous reviewer for pointing out this issue.

The authors declare that they have no competing interests.

Daniel Graziotin conceived and designed the experiments, performed the experiments, analyzed the data, contributed reagents/materials/analysis tools, wrote the paper, prepared figures and/or tables, reviewed drafts of the paper.

Xiaofeng Wang and Pekka Abrahamsson conceived and designed the experiments, performed the experiments, analyzed the data, contributed reagents/materials/analysis tools, wrote the paper, reviewed drafts of the paper.

The following information was supplied relating to ethical approvals (i.e., approving body and any reference numbers):

Institutional review board approval for conducting empirical studies on human participants was not required by the institution. However, written consent was obtained from all of the subjects. The participants were advised, both informally and on the consent form, about the data retained and that anonymity was fully ensured. No sensitive data were collected in this study. The participants were assigned a random participant code to link the gathered data. The code was in no way linked to any information that would reveal a participant’s identity.

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
