# Peer review of "Happy software developers solve problems better: psychological measurements in empirical software engineering"

_PeerJ, doi:10.7717/peerj.289_

## Round 0.1 · original submission · Major Revisions

The paper has been reviewed by two independent reviewers who have made extensive and detailed comments about the current ms. Both pointed out many serious issues with the paper, and although one suggested rejection, I reviewed closely the policies of PeerJ and feel that the basic research upon which this is based is well-conceived, and the comments are mostly about presentation, wording, and (critically) placing the work appropriately within existing research in I/O psychology. For this reason, I have decided to return it for major revisions, during which I hope you address or respond to all of the substantive comments made by reviewers.

At this point, I will not give detailed feedback as I expect the manuscript to change substantially in the next round. I do expect to have a number of suggestions/edits to the figures and tables, and the statistical analyses. In general, these do not appear to be presented entirely clearly, or in standard ways. Here are several major issues you should focus on:



1. At a conceptual level the writing is fairly adequate, but at an execution level it needs to be seriously improved. Many detailed examples are given by the reviewers, but you should make special efforts to improve its readability, over and above the specific comments made by the reviewers. One tip I'll give at this point is a lot of 'glue' verbiage exists telling the reader what the section is supposed to be about. This is sometimes useful, but it should be apparent to the reader by the section headers, and I think it should be avoided. Anyway, I won't send a revision out for re-review until I'm happy with the its readability.

2. I understand that for a computer science-focused project, it is a substantial bridge moving into behavioral validation methods. Within psychology field, 'Psychometrics' is not a label you can use just because you used behavioral methods; I would expect detailed analysis of the psychometric properties of the test, including things like test-retest reliability, maybe some type of split-half analysis, perhaps analysis via item response theory to examine the importance of different questions, etc., as well as perhaps some measurement of criterial validity or some other validity assessments. Within software engineering, however, I can see how this might be considered a 'psycho-metric', in comparison to other metrics an engineer would consider, but its use is misleading from an empirical science perspective. You should make an effort to either establish the psychometric properties of the tests, or choose another term, downplaying your (mis)-use of the term psychometrics. Perhaps a 'behavoiral metric', or something like that, or making the title "an empirical approach".

3. Along the same line, greater effort should be made to place this work within the context of I/O psychology; many pointers were given by the reviewers, and you should incorporate this to the extent it is relevant to your approach.

Reviewer 1 ·

Basic reporting

Throughout the manuscript, there were numerous grammatical errors and awkwardly worded sentences that made the manuscript slow and more difficult to read. Furthermore, there were numerous cases in which entirely incorrect words were used. For example, line 309 states, “Participants were advised (both informally and on the consensus form)…” This sentence should read “Participants were advised (both informally and on the consent form)…” The words consensus and consent mean entirely different things. The authors should have someone with a strong English background edit the manuscript to eliminate these mistakes throughout the manuscript.

Introduction and Related Work Sections:
The researchers make an admirable attempt to bridge the gap between the field of computer science and the field of psychology. However, the introduction does not provide a solid theoretical background or basis for the study. The background provided for the role that affect plays in job performance was significantly lacking. For example, Affective Events Theory (AET) is a theory used to explain how moods and emotions influence job performance and job satisfaction. The authors should consider the inclusion of AET as a theoretical basis for their research. Furthermore, the authors would be advised to review the research in the field of Industrial/Organizational Psychology, where much work has already been conducted examining the role that affect, emotions, and moods play in job performance and various job related outcomes.

For more information on AET see:
Wiess, H. M. & Cropanzano, R. (1996). Affective events theory: A theoretical discussion of the structure, causes and consequences of affective experiences at work. In B. M. Staw & L. L. Cummings (Eds.), Research in Organizational Behavior: An Annual Series of Analytical Essays and Critical Reviews (p. 1-74). Greenwich, CT: JAI Press.

In addition to lacking a strong theoretical foundation, this manuscript consistently used the term psychometrics incorrectly. Psychometrics is a subfield of psychology that focuses on the theory and techniques of psychological measurements. More specifically, the field of psychometrics deals with the design, development and validation of psychological measures. Throughout the manuscript it seemed as if psychometrics was being used as a “buzzword” to excite the reader(s) and was being used in reference to simply the use of a psychological measures or some kind of benefit that is provided by using a psychological measure. This is not correct.

In general, the Introduction section and the Related Work section both seemed very disorganized. The flow of the manuscript did not create a logical or theoretical progression to the hypotheses. The incorporation of a theoretical foundation, such as AET, should help to better develop the rationale for the hypotheses.

In the Related Work section, much of the literature cited from the field of psychology is outdated or entirely false. For example, the authors state, “It is difficult to differentiate terms like affective state, emotion, and mood.” However, much of the research in the field of social psychology agrees that emotions, moods, and affect are entirely separable constructs. Therefore, the authors should review more recent literature dealing with affect, emotion and mood.
Constructs and Psychometrics Section:
If the authors are not going to use the PANAS (as stated in lines 212 – 220) do not include background information about the PANAS.

When discussing the PANAS, the authors cite that the PANAS only captures high-arousal feelings in general. This statement is only partially true. There are different versions of the PANAS depending on the level of affect the researcher is interested in studying. For example, the PANAS can be used to measure more stable trait level affect or more momentary state level affect.

When discussing the SPANE, the authors state that the SPANE assesses positive and negative experiences during the past 4 weeks. If the researchers are truly interested in how affective states influence job related outcomes, the researchers should not use a measure that aggregates experiences across 4 weeks. A scale that measures positive and negative experiences that occur over 4 weeks is not measuring affective states. Although the authors cite reasons for not using the PANAS, the PANAS would have been better suited to measure affective states because one version of the PANAS asks participants to rate the degree to which they are currently experiencing various positive and negative feelings. Thus, the PANAS would have been able to actually measure changes in the participants’ actual affective states.

Similarly, when discussing the SPANE, the authors state that the SPANE-B could be used to split participants into groups using a median split. This is a very arbitrary and inappropriate technique for several reasons. First, median splits artificially create groups. Second, median splits limit the types of analyses that can be conducted. Third, similar to the second, median splits remove precision that was originally available in the continuous measure.

Experimental design

Hypotheses Section:
Lastly, when the hypotheses are presented it is not clear exactly what they research is examining. The hypotheses should be clearly stated in words not formulas. By not clearly stating the hypotheses in words, it is confusing to the reader what is being examined. For example, without flipping back pages, it is unclear what BCRn-pos actually means and how it is different from BCRpos. In a related comment, the authors do not actually define and operationalize what pos and n-pos actually mean.

Context and Participants Section:
The researchers state that a “formal experiment” was conducted. However, given the information that is presented in the manuscript, this is far from true. A form experiment is a study in which there is random assignment to groups and the researcher actually manipulates the independent variable. Neither of these things was conducted. The authors could make the argument that a quasi-experimental design was used. However, a correlational research study best describes the research methodology employed in this study.

The information provided in the Participants Subsection of the Results and Discussion Section (lines 395-405) should be moved to the Context and Participants section. There is no need to have two Participants sections in the paper.

Procedure Section:
Lines 356 – 361 should be removed. Instead the authors should simply state what the alphas were for the various measures.

Lines 370 – 379 should be removed. They detract from the flow of the paper.

Lines 382 – 383: There were no conditions in the study. No variables were manipulated.

Validity of the findings

Hypothesis Testing Section:
The first thing that is stated in this section is the “the data could not permit insightful regression analysis.” This is entirely false. The data could have been analyzed using regression analysis. In fact, the data would have been more appropriately analyzed using regression analysis for multiple reasons. This would have allowed the researchers to retain the precision of the SPANE-B as a continuous scale (without dichotomizing the scale using a median split). Second, the regression analysis allows for the researcher to control for theoretically or empirically confounding variables. Third, regression analysis does allow for the test of curvilinear relationships. This last point is mentioned because Figure 8 and Figure 9 appear to display a curvilinear relationship. Lastly, regression analysis is fairly robust against minor violations of normality.

The presentation of the inferential statistics is inappropriately presented. When presenting psychological research, one needs to adhere to the guidelines in the Publication Manual of the American Psychological Association 6th Edition.

Evaluation of Results and Implications Section:
The conclusions drawn from this study are not entirely supported by the data. For example, line 470 states, “The empirical data does not support a difference in the creativity with respect to the affective states of the participants.” The distribution of scores shows that most people were positive in their affective states. Additionally, the SPANE-B does not really measure affective states. Similarly, lines 478 – 482 are not supported by the data or results. First, no analyses were conducted examining the degree of relationship between ACR and SPANE-B scores. Therefore, lines 478 – 479 cannot be stated. Lines 481 – 481 state that specific relationship cannot be claimed at this stage. This is also false. Incorrect data analytic procedures were used in this study. If multiple regression techniques were used, specific relationships could actually be determined.

In line 500, the manuscript states, “A paired t-test proved that the increase…” This is scientifically inaccurate. In research, we never prove a hypothesis. We only support or fail to support a hypothesis. Proof implies indisputable evidence.

Lines 514 – 516 are also not supported by the study or the data. First, no mood induction techniques were actually used. The researchers manipulated none of the participants’ moods. This study could not have empirically demonstrated that the SPANE is capable of detecting mood induction effects because no moods were induced.

Validity of the Study Section:
Line 533 states, “Although the number of the participants is acceptable…” How was this determined? There was not any discussion regarding an a priori power analysis.

Reviewer 2 ·

Basic reporting

The structure of the article is not according to PeerJ template structure.

Abstract-Page 1
Abstract is bit lengthy. I think there is no need to provide justification of the study in the abstract.

Introduction
Page 3
Line 45: There should be a dot “.” after reference. If this is one sentence i.e. from line 44-Line 48, “Cognitive processing abilities … problem solving skills” then this sentence should be broken down into two or more sentences for clarity.
Line 48-50. Need citation for the claim

Page 4
Line 67: “from the point of view of researcher” Why is it from the point of view of researcher? It is not clear.
Lines 70-73: shows conclusions which will be better if put in the conclusions section

Page 5
Line 118: By converting “In this the discrete approach…” to “In this article/study the discrete approach…” will make it clearer
Lines 122-131: could be grouped under the heading “Mood Measurement” or similar heading. This will make paper more clear and readable

Page 7
Line 215: Repeated “David Watson, Clark, & Tellegan, 1988)”

Lines 216-218: Sentence need to be rewritten

There is no clear reason and literature support that explain authors’ decision of not considering PANAS and why they opted for SPANE. Some discussion about which affects measurement instruments SPANE converges and how SPANE would be worthwhile as compared to other affective instruments might further clarify the subject under consideration

Under Heading Constructs and Psychometrics, first two paragraphs discussed selection of the affect measurement instrument while next four paragraphs discussed about selection of creativity tasks. The last two paragraphs discussed selection of Analytical Problem Solving and its related tasks. Grouping these paragraphs under some suitable sub headings will improve readability of the paper

Experimental design

Page 9
Lines 298-300: I think there is no need to write research and experimentation guidelines in the paper. These guidelines are to follow. Instead authors can write general summary of what is coming in the next section.

Line 282: “which derive…” should be changed to “which are derived ….”

Lines 283-284: “The alternative hypotheses are two tailed because the literature review does not provide clear indications” This sentence is not clear. Of what the literature review is not providing clear indication. As literature is reviewed by authors, they could provide this indication.

Line 266: Change “Hypothesis H1, H2 …” to “Hypotheses H1, H2 …”

Lines 291, 292, 293: What N-POS and POS stand for. I am for now taking N-POS for developers feeling non positive affective states. But this still keep a doubt in reader mind. Therefore a clarification for abbreviation is necessary.
Line 312 Now = No

Page 10
Line 324: What is Article S1. There is no attached information about it.
Page 11
Lines 370-379: The instruction about a particular task to the participants should come at the point where task are being explained. For example creative task instruction sentence should come at Page 10, somewhere between lines 334-339.

Page 12
I think there is no need of detailed descriptive statistics. Some important statistics can be highlighted under the heading “hypotheses testing”. In addition, there is no need of the box plot figures of non-significant difference hypotheses. There is also no need of Figure 6 and Figure 7.

Validity of the findings

No Comments

Additional comments

In general a research writing should be in the past tense. Therefore, it is advised to authors to use past tense in the paper and especially use past tense in experimental design and results section

---

## Round 0.2 · Major Revisions

Daniel,

I have reviewed the manuscript in detail, and feel that it still needs substantial revisions. These primarily relate to two aspects of the writing--the English needs improving in general, and it should conform better to the norms of reporting behavioral/psychological research. Substantively, there are a few major issues: 1. You should conduct and report a regression analysis establishing whether more extreme affective state scores really produce higher outcome scores (or just remove all the discussion of this), 2. The manuscript needs to be streamlined, especially the introduction. I understand the desire to establish links to past literature, but much of this is not germaine to the empirical question at hand, and you should make every attempt to remove irrelevant information, even if it is factual. In its current format, I won't send it back to any reviewers because I think it needs to be much more readable before it can be fairly evaluated.

I have provided detailed line-by-line feedback below, including many wording suggestions. Addressing these wording issues will be necessary, but not sufficient, for acceptance, and you should treat them as a starting point for improving the paper.

I'm going to return this for major revisions, and if the revision meets two criteria, I will return it to at least one reviewer for a second opinion. 1. The writing must be improved to a standard typical for an English-language psychology journal (you should have a native English speaker proof your revision before resubmitting. PeerJ may have resources to support this). 2. It must be shorter, not longer, than the current version.

That said, I think the basic results are interesting, and add to the currently mixed literature on the topic. I would not have invested the time I have in making the comments below if I did not feel this is worthy of publication. I encourage you to make every effort to improve the manuscript, as I believe doing so will greatly affect its ultimate impact.

Best,
Shane



Detailed Comments


Opening paragraph:

- it is not clear what the 'productivity gap' really refers to. I'd suggest defining it in the first sentence. I think the revisions here in the intro make this less clear than the abstract does--the so-called productivity gap (whether or not it exists) seems to be that the NEED for software is increasing faster than the industry's ability to deliver it (rather than the amount of software).

I'd also take issue with the notion of the productivity gap. One look at any software repository or store shows that there is no lack of software, and it is likely that most of the software out there doesn't even pay for the developer's time spent writing it. But in terms of usable, useful, effective, robust software, this is probably true.


But I don't think this is really important for the present article; the research is sound regardless of whether a gap exists. Maybe an equally-good justification comes from the study of software staffing, such as the mythical man-month, and related work that establishes that in general, software projects are typically late, over-budget, and do not deliver what is promised?


Either way, if you want to keep the current introduction, I'd suggest wording such as:
For several years, the software industry has been faced with the problem of the productivity
gap (Ecker, Müller, & Domer, 2009): The need for software to operate within electronic devices (including personal computers, mobile phones, televisions, automobiles, and many others) is outstripping the software industry's ability to deliver it....

line 12: "while there have been several solutions proposed over time"; I'd suggest
"Although solutions have been proposed,"
Line 13: 'exists' would be better than 'pertains'

Line 15: (e.g., Bohm & Papaccio, 1988). Also, this statement needs to be clarified. What does it mean to focus on 'people'. I assume that it means something like "focusing on the selection, training, support and happiness of software developers"

Line 16: 'in specific' -> Specifically,

Line 17: If the quotation is a direct quote from Cockburn & Highsmith, it should be stated as such from the outset (As noted by Cockburn & Highsmith (2001), "If the people..."

Line 22: challenge, when compared -> 'challenge in comparison to'

Line 23: I'd suggest 'engineering fields: software development is substantially more complex than industrial processes ....'

Line 24: 'fascinating' is a subjective assessment that should be avoided.

Line 26: Lately -> Recently
Line 28: 'A proposal to satisfy' -> 'For example, Feldt et al. (2008) proposed that the human aspects of software engineering could be studied empirically, by "collecting psychometrics"1.


Line 30: "this call has been mildly echoed" -> 'although this proposal has begun to gain traction, limited research has been done specifically on the role of emotion and mood on software developer productivity'

Line 31: This is the first time emotion and mood has been mentioned. I'd suggest perhaps adding it to the first paragraph when describing what you mean about 'focusing on people'


Line 33-35: The statement needs to be sourced, and I have a hard time understanding what it is trying to say.

Line 35-37: If there are numerous studies, you should cite more than 1. Suggest changing 'happiness of people is an indicator of' -> 'reported happiness is related to achievment'

38: sense-making -> sensemaking (one word)

39. The statement about 'organizational settings' does not fit here--you should discuss it within the role of emotion, or get rid of the statement.


42: accelerated -> increased

46: like -> such as

48: Despite 'the fact that'
50: 'is lacking an understanding...' -> lacks an understanding of the role of emotions in the software development process


Footnote: How about: "The software engineering literature has sometimes used the term 'psychometrics' to describe general psychological measures that might be used along with other software development metrics. However, 'psychometrics' has very a specific meaning within psychological research, involving establishing the reliability and validity of a psychological measurement. In this article, we use the more appropriate 'Psychological Measurement' to refer to this concept.


55. I think saying that it is necessary is yet to be established. How about 'Thus, we believe that studying affective states of software developers may provide new insights about ways to improve overall productivity'

57. You can't really know whether software developers are mostly doing problems solving; maybe say "Many of the tasks software developers engage in require problem solving. For example, they need to..."

61: Both of 'these'

61-62: not physical -> typically not physical

64: "Some cognitive processes have been shown..." (Many cognitive researchers would take issue with the assumption that the cognitive processes are all, or even usually, linked strongly to emotional state)

65-71. Do you have a reference for research that has studied the correlation, and that no consensus has formed? If not, I'd suggest deleting from 'Research in psychology' to 'job-related performance', and start the last sentence with "Furthermore, to the best of our knowledge, the relationship between ...."

72-76: This paragraph can be deleted, and perhaps moved/adapted to an experimental method section and the results. For research like this, it is atypical to describe the results before you describe the method of the study. This section should introduce and motivate your experiment.


83-92: I don't think there is a need to outline this paper, as it is fairly straightforward. Maybe simply say at the end of the previous paragraph, "Next, we will review some of the background research on how affective state impacts creative problem solving. Following that, we will report a new experiment that establishes the relationship between affect and productivity in software developers.

93-162. Although I understand the reason for this section, and the motivation is clear in the footnote, I think it can be tightened up substantially. It is certainly true that emotions, mood, feelings, are used interchangeably and can be defined more precisely. But this does not seem to be the purpose of the present paper, and spending time in definitions is a distraction.

There is value to defining and justifying exactly what you are measuring. But why is it important to know that there are two theories used to categorize affective states? Unless the present study attempts to support one theory over the other? I think this entire section could be reduced to about a paragraph that has a laser focus, describing the particular aspects of emotion you care about. More important is to motivate the ways you want to measure affective state (the next section).


Here are some wording suggestions within this section, but I would prefer you cut it down to about 2-3 paragraphs.
93: Maybe preface this section with the last sentence in the first paragraph of the page. "In daily conversations, the terms emotions, moods, feelings, and happiness are freely interchanged. Thus it is useful to define these terms. In general, _affective states_ have been referred to as 'any type of .....'
94 I don't think you need bracket around the elipses, just four dots ....
95: "However, it has been employed more generally to mean"

103: often defined -> 'typically defined'

105: maybe 'valenced emotional states in which the individual can feel either good or bad'

109: No need to cite Ed or E. diener unless there is a chance of confusion in the reference list.

112: "outside the field of psychology"

114: consider -> have considered

117: For example, it has been suggested...


164: grouped participants... -> 'categorized affective states in terms of negative, (occasionally) neutral, and positive valence'

165: 'grouping is often based on manipulations that induce affective state'

166: Instead of 'In the case of quasi-experiments... "Alternately, some studies have used quasi-experimental designs that select participants having different affective states, usually based on answers to questionairres". This should be moved to the end of the paragraph.

171: Recent studies -> "However, recent..."


175: (PANAS; Watson, Clark...)

176: "However, several criticisms have been made about this instrument"

179: 'Another limitation... -> "Others have argued that it is not sensitive to or inclusive of cultural differences in the desirability of emotions (citations...)

184, Recent, modern... -> "Recently, scales have been proposed that reduce..."

188: divided into two subscales (SPANE-P and SPANE-N) that assess positive and negative affective states.


194: change to -> "six items, and thus range from 6 to 30"


209: unclear what 'proved consistency' means; maybe 'provided consistency' or 'has proved to be consistent'

211. I don't think SPANE-B is 'fuzzy' in any sense, but it does provide graded scale rather than a categorical scale.

214. I don't see any need to say regression analysis is possible here. Regression analysis is always possible.

216: Cronbach's alpha need not be introduced here, unless you want to cite previous research that measured alpha on the SPANE. But you don't cite anything; if you use this, it can be briefly mentioned in the results section.


221.Maybe just state something like that "Several past research efforts have examined the role of affective state on software developers. For example, Shaw (2004) observed..."

226. The sentence 'however, little or no attention' is somewhat redundant, given you said it before, and somewhat incorrect, given you just cited somebody who did this research. I'd recommend deleting this entire sentence.

238: No need to include the second-to-last sentence of this paragraph.

239: Shaw (2004) concluded by calling for additional research....


The entire paragraph 241-255 is probably too detailed. Maybe (after an intro sentence such as "This call was answered by Khan et al (2010), who found a correlation between cognitive ability and software development skill", you could skip directly to the last half of the paragraph.

256: "Finally, Graziotin ...."

258: 'and included a pictorial survey'

264: I don't see need for saying that you previously called for additional research on the topic you were studying.


Section: Problem solving skills and affective states.
This section should do a better job of setting up your hypotheses, and begin to engage in 'convergent thinking'. I don't believe the convergent/divergent dichotomy is necessary to discuss here, as you end up using other terms (creativity and analytic problem solving), and don't really refer to it again. You should try to use 2 sentences to define convergent/divergent and their rough mapping onto creativity/analytic problem solving, but then move directly into discussing how affect might influence these two modes of problem solving, that starts on 282.

For example:

"Researchers have sometimes distinguished between two modes of creative and analytic problem solving: convergent and divergent thinking (Csiksze....), which map roughly onto creativity and analytic problem solving.

272: already known. -> already known, and leaves little room...


276 analyze-> examine

278-279: This paragraph is a non sequiter...this statement could be made later, in the method section, where you describe your creativity measures. Instead, you should continue the argument made in the previous paragraph about problem-solving styles, and make a direct connection to affect. This paragraph does not, and
279: no T..M. needed in Amabile citation

282: This paragraph needs a topic statement that summarizes the argument up-front. Something like, "Research has found relationships between positive or negative affective state and problem solving.' Also, the results don't appear to be completely consistent, so maybe 'Past research has found mixed evidence regarding the relationships between positive or negative affective state and problem solving'


283: delete 'in terms of the quality of generated ideas'; change 'higher creativity' to 'higher quality of generated ideas'


287: with respect to -> in comparison to (same for line 288)

288: you don't say which direction the effect of positive affect is.

289: If the Lewis paper follows from the previous, you should connect them somehow; 'Similarly, Lewis et al. (2011)....

290 with respect to -> in comparison to

295: there are studies empirically demonstrating -> studies have demonstrated

296: unclear who you are referring to as 'authors of this study'. The last few sentences could be 'As argued by Fong (2006), no clear relationship has been established between affective state and problem solving creativity.'


303: this paragraph could be moved before the statement of the hypotheses (see next comment), with a better transition from the previous paragraph. e.g., "In contrast to the case for creativity, fewer studies have investigated how affective state influences analytic problem solving".



301/314. I'd prefer if you simply state your hypotheses in text, rather than labeling them 'first high-level hypothesis'. E.g., "Because of the lack of a clear relationship between affective state and creativity exists, we designed an experiment to test two related high-level hypotheses. We hypothesize that affective state will impact (1) the creative work produced by software developers and (2) their analytic problem-solving skills." This could lead directly into the paragraph/section labeled 'hypotheses' (and either move that header to the beginning of this, or get rid of it entirely), as the lines 316-365 might better be placed in the 'Method' section.


367-372. If you move these next to the statement of the high-level hypothesis, you can get rid of these entire paragraph.
373-383. The statement of the sub-hypotheses is really hard to follow.I don't see a fundamental difference between H1/H2/H3; these seem to be simply three measures that can be used to assess hypothesis 1 I'd recommend deleting the specific statements of these hypotheses, as it is time to start describing the experimental methods. These can be folded into a results section, as measures used to assess/test your hypotheses.

316-365. As much of this as possible should be moved into the This should be moved into the Materials and Methods section and the results section. and streamlined to focus on what you did in particular. Then, distill the high-level description down to one or two paragraphs that lead you directly into the experiment.
The detailed description of measures (e.g., lines 347-353) could go directly into the method if they are not redundant.


* First say that to test the hypotheses, you needed to develop or obtain measures of each oof the two problem solving skills.

*Then, give 1-2 sentences about assessing creativity, citing the references you did.
* Say that "we used a 'caption-generating task' to assess creativity, and used subjective ratings to determine the quality of the captions (as well as recording the quantity of creative output).


Now, in a second paragraph, say 'to test analytic problem solving, we used the Tower of London (Shallice, 1982), a game designed to assess planning and analytic problem solving.' You can add some of the discussion and justification here, but the description of the measures does not belong in the introduction.

The last paragraph here (354-365) should be the last paragraph before your method section.


354: To the knowledge of the authors of this article -> to our knowledge
355: that define -> using
360: I think the statement on 357-359 is clear enough that no more detailed example is necessary.

364: presented -> present
364: Consequently, creativity and ....


Materials and methods (Note that some of the material in the previous section shoul dbe moved here)

388: what are 'credit points'?


391: provenience country -> 'nationality'
393: has been -> was

398: has been -> was

402: experiment-> experimental
402: delete 'working'

406. This section is really about data analysis, not participants, and should be moved to the end of the materials and methods section. This will make it easier to discuss, because you will already have described the task.
407-408: I don't think the citations are needed; it is unclear what they indicate.

408-410. How about "After the data were collected, ten independent judges were contacted to rate the captions produced in the creativity task, seven of whom responded and five of whom completed evaluation of the captions. These five included two professors of Design & Arts, two professors of humanistic studies, and one professor of creative writing." If you move this later, you can dispence with the sentences on lines 413-415, and move directly into "The judges were blind to the design and scope of the experiment, and rated the captions independently.

423:Say something like "A version of the Tower of London task implemented in the Psychology Experiment Building Language (Mueller, 2012) previously used to examine age-related changes in planning and executive function (Piper et al., 2011) was used to assess analytic problem solving" The Mueller 2012 should cite the version of the software you used (i.e., PEBL Version 0.13). In addition, there is a paper currently in press at J Neuroscience Methods that will be appropriate to cite if it is ready by the time your paper is accepted (tentatively Mueller & Piper, 2013)

431: affective states measurement (SPANE)

451: Were the instructions in English, or were they translated into Italian?

458: I don't understand the intent of the sentence starting 'This apparent freedom..."

465: 1. You don't need to described data analysis techniques in the method section.
2. To cite R in publications use:

R Core Team (2013). R: A language and environment for statistical
computing. R Foundation for Statistical Computing, Vienna, Austria.
URL http://www.R-project.org/.


467-470. This should be moved to the procedure section. There is no need to call this a 'natural' experiment, which is a term I've only seen in textbooks; perhaps only refer to it as a 'study'.

*Note that some of the discussion of data coding could be moved here.
472-73: This belongs in the method, not the results.

*In the results section, for every t-test, please report an effect size (e.g., Cohen's d). For the Mann-whitney/wilcox test, Rand wilcox has developed an effect size measure that has similar assumptions, but it can be hard to compute, so at least report Cohen's d.

474: You should remind the reader of what a SPANE-B of nine indicates. Also, the split-half method may make it appear you are comparing people with positive emotion to those with negative, but I think you are really comparing high positives to low positives/neutral with five negatives.

475: no paragraph break.

476: It is unclear why regression would not provide a meaningful analysis. It seems like a 2nd order polynomial regression would be perfect for establishing the curvature.

492: State the specific hypotheses, rather than referring back to a coded 'H1'



492-504: You have already put the means and confidence intervals in a table, so no need to repeat in text.

493: Report the t-tests something like this: 'There was no significant difference between N-Pos and POS groups on the BRC score (t(39)= .2...; or the ACR score (t()=XX).'

497: restate H3, and use the pattern above to report the results.

503: it seems strange to reconfigure your groups based on repeated testing. Why not use the average score on both SPANE measures to create one single grouping? How many participants were recategorized? I see this is discussed somewhat in line 554-562. Maybe that discussion shuold be moved to this point, as it does not really directly bear on the hypotheses.

505-513: Visual inspection of distributions of small numbers of observations can be misleading. You should just report the results of the appropriate test, rather than reading tea leaves.


You should provide a statistical model testing the slope of the relationship within npos and pos groups. This might be two separate regressions, or it might be a single regression with a polynomial.

525: in terms of any of the creativity measures we used.

528: If they were not significantly in line with the finding, they were not in line with the finding. You might say 'we found no significant difference in the number of creative ideas generated, in contrast to Sowden and Dawson, who found that ....."


533: Without the regression analysis suggested earlier, this paragraph is just speculation and is unwarranted. It is not clear that you had SPANE-B scores approaching those extremes, especially the negative (lowest was about -9)


541: The conclusion that 'happy' developers are better analytic problem solvers in incorrect. Almost all the developers were 'happy' according to the scale. Rather, it shows the happiest developers were more productive than the less happy developers.

544: again, you didn't test many with negative affective states, you compared happiest to moderately neutral. Consequently, this result is not a contradiction.


547: As discussed earlier, run a statistical test regarding this u-shaped function, or don't bring it up. A quadratic regression will produce a low point that you can report specifically.


551: this paragraph should be stated much earlier, when you discuss the split-half method.


Implications: it seems that the results have implications for management styles and may partially justify the 'lavish' workplaces present in silicon valley. However, if the results were to generalize, we would suspect that creative problem solving won't be impacted in general, but analytic skills might be.


References: several references used first initials when citing them in-text. This should be avoided.

---

## Round 0.3 · Minor Revisions

I've sent the revision out to one of the reviewers who reviewed an earlier version, and he is happy with the current version, but makes a couple suggestions. These sound very reasonable, and I think you should try to address them. Because the semester just started and I have a backlog of things in my schedule, I don't have time right now to do a detailed read of your latest revision for another week or so (I've skimmed it and your rebuttal document, and see that you have dealt with my comments). My dilemma is to either make you wait until I can review in more detail, or send this back right away to give you a chance to address the reviewer. I've opted for the second, and when you complete this (which I don't anticipate taking more than a couple days if you have the time available), I'll do a final readthrough, and will either accept it as-is or give some additional comments which I anticipate would be limited to wording issues and the like.

Reviewer 1 ·

Basic reporting

Authors have improved paper and its writing. It’s now easier to read and understand the paper. Couple of discrepancies I was still able to find are:

On Page 13 from line 462-466, authors discussed results of this research in terms of analytical problem skills contradict with researches like Abele-Brehm (1992), Melton (1995) and Kaufmann & Vosburg (1997) etc. Couple of lines of whether this research agrees with some previous researches will add value to the paper. Consequently a descriptive paragraph for reasons of contradiction might prove to be of a higher value.

Experimental design

OK

Validity of the findings

I think the article need a discussions and need to point out some reasons why the results are in contradiction if the opposite findings is a norm. Else they need to cite some research which agrees with these findings.

---

## Round 0.4 · accepted · Accept

Here are a few minor wording changes you should make during the copediting process:

line 241-243: Not sure what this is trying to say. Maybe "Consequently, based on the limited studies, no clear prediction about the relationship between affective state and analytic problem solving skill can be made"

305: the M&P paper is 2014
413: Check that 'Wilk' is correct in 'Shapiro-Wilk'; I think it might be Wilks or Wilkes

421: Cronbach (1951) developed the alpha coefficient (second Cronbach is redundant). If possible, probably should use greek alpha here.

428: suggested -> suggest

438: double-subscripts are a bit awkward here (maybe elsewhere too; line 508)

544-> delete 'ever'